# Plantar Pressure Distribution in Female Olympic-Style Weightlifters

**DOI:** 10.3390/ijerph17082669

**Published:** 2020-04-13

**Authors:** Arletta Hawrylak, Hanna Gronowska

**Affiliations:** 1Department of Physiotherapy, University School of Physical Education, al. Ignacego Jana Paderewskiego 35, 51-612 Wroclaw, Poland; 2Physiotherapist, Private Practice, P.A. Hlonda 10/121, 02-972 Warszawa, Poland; hanna.gronowska@gmail.com

**Keywords:** women, weightlifting, baropodometry, biomechanics, training

## Abstract

*Background*: The aim of this study was to investigate differences in static and dynamic plantar pressure and other foot geometry variables between female Olympic-style weightlifters and an age- and sex-matched control group. *Methods:* The study involved 24 national-level competitive weightlifters and 24 physical education students. Leg dominance was determined and baropodometry was used to assess plantar pressure and foot posture during quiet standing and walking. For all variables basic descriptive statistics were calculated (mean ± standard deviation). Student’s *t* test was used to compare the between-group differences. Spearman’s rank correlation coefficients were calculated to determine the association between BMI (Body Mass Index) and average and peak plantar pressure. *Results*: No significant differences were observed in plantar pressure distribution between the two groups. A moderate correlation was found between BMI and non-dominant limb peak and average plantar pressure in the group of weightlifters in the static condition and in the control group in the dynamic condition. *Conclusions*: Olympic-style weightlifting may affect plantar pressure distribution and foot posture in female weightlifters. BMI may also correlate with peak and average plantar pressure in this population. Further research is needed to determine if Olympic-style weightlifting may affect plantar pressure distribution and foot posture in female weightlifters.

## 1. Introduction

Professional involvement in sports is associated with overtraining, which can adversely affect the human skeletal system. Athletes are at risk of chronic injuries and overuse syndrome induced by excessive training loads or the repetitive use of specific muscle groups, the most common being the knee joint (28%) and ankle and knee joints (21%). Disorders of the feet are one of the factors that can increase the risk of injury [1]. The highly intensive nature of weightlifting training and competition can induce numerous structural and functional adaptations that include, among others, to the musculoskeletal system [2]. Until recently, weightlifting was considered a discipline conducive to spinal injury [3]. Among weight-lifters, injury is most often experienced at the knee joints, spine, and radiocarpal joint. Performance in Olympic-style weightlifting is multifactorial with no single variable a predictor of success [4]. Particular attention should be paid to understanding the mechanisms of adaptation that women undergo in this discipline, as the majority of studies have involved only male athletes. Women competed for the first time in the world weightlifting championships in 1987 and in the Olympic Games in Sydney in 2000 [2,5]. The number of female participants continually increases due to changes in social perception but also as a result of increased knowledge of the benefits of weightlifting on enhanced performance, health, and reducing musculoskeletal injury [6].

Female Olympic-style weightlifting involves the snatch and the clean and jerk. Both require a very high level of coordination and proper technique during execution [2,7,8]. The snatch starts from a squatting position and involves lifting the barbell over the head in one dynamic movement. The clean and jerk consists of two phases: lifting the barbell to the chest and then lifting the barbell above the head. In this lift, the weightlifter stands up from a squat to a stable position and then, in one dynamic movement, lifts the barbell above the head at which point the legs are split into a lunge position [8]. Even seemingly minor errors such as a momentarily loss of stability, muscle imbalance, or inadequate concentration can lead to failure [2,8]. Maximizing the synchrony of the entire kinematic chain in weightlifting is critical for optimal performance as it involves the coordinated activation of the entire body [5]. The feet are a particularly important component in lift performance as they provide a dynamic base of support throughout the entire lift [6]. Proper foot placement in the starting position and later phases of the lift dictate maximal force production, lift stability, and overall movement efficiency [2,9,10]. Several studies of weightlifting athletes have focused on squat technique as a method for increasing lower limb strength [11,12]. However, there is a lack of research on the effects of weight training on foot structure and plantar pressure distribution in athletes. Foot structure including arch height and foot shape can affect whole-body balance and stability in a wide range of situations, and previous studies have found that even amateur involvement in strength sports can modify the morphological structure of the feet [13]. Excess forefoot loading with pronation may cause significant metatarsal and plantar aponeurosis pain [14]. As the feet provide a base of support during lift execution, assessing their functional status in both static and dynamic conditions can identify functional asymmetries or weaknesses in the biokinetic chain of an individual [15]. Signs of disrupted plantar muscle tone may indicate reduced hamstring muscle length, loss of normal lordotic curvature, or hyperextension injuries of the cervical spine [15,16]. As the highly intensive nature of weightlifting training and competition is known to induce several functional and structural adaptations particularly in the musculoskeletal system, it is possible that weightlifting footwear might also induce several changes in plantar structure and loading [2]. By enhancing lift biomechanics and providing additional support, weightlifting shoes are credited with facilitating movement execution and technique as well as preventing injury [17]. The most important design element of weightlifting shoes is a stiff and inflexible sole that does not compress, allowing a consistent transfer of force and improved stability. Another is an elevated heel, which limits the need for hyper-dorsiflexion during the squat phase of each lift, allowing for a more upright torso that reduces loading on the spine and hip joints [18]. The raised heel of weightlifting shoes also allows for a neutral pelvic position that further enables postural stability [19]. Additionally, weightlifting shoes have been found to increase muscle activation and involvement of the lower extremities [17]. The relatively late introduction of women in this sport warrants a reexamination of the structural and functional adaptations as the majority of existing research involved only males [4]. Therefore, the aim of this study was to determine if female Olympic-style weightlifters show differences in foot shape and selected plantar variables in both static and dynamic conditions compared with an untrained control group. The study was designed to answer the following questions: Does any type of foot shape prevail among female weightlifters (neutral, pronation, supination)? Are there significant differences in the distribution of plantar pressure and plantar loading symmetry between female weightlifters and an untrained cohort? Is there a correlation between BMI (Body Mass Index) and average and maximum plantar pressure in both groups?

## 2. Materials and Methods

### 2.1. Experimental Approach to the Problem

Baropodometric analysis was used to assess foot posture and plantar pressure in dynamic and static conditions in female weightlifters and age- and sex-matched controls.

### 2.2. Subjects

The weightlifting group involved 24 female participants with a minimum two years of experience in competitive weightlifting selected during an Under-23 (years of age) national championship. This group trained 4–6 times per week with each training session lasting 120 min. The control group consisted of 24 female students of the Department of Physiotherapy at a physical education university. They declared that they were not involved in any sport or did not perform strenuous physical activity. All were free of any musculoskeletal injury or dysfunction and provided written informed consent to participate in the study. Basic anthropometric characteristics for both groups are presented in Table 1. The study was performed in a research laboratory belonging to the Department of Physiotherapy of the University of Physical Education located in Wrocław, Poland. The study was conducted in compliance with the Declaration of Helsinki and was approved by the local ethics committee. The design was approved by the Senate Ethics Committee for Scientific Research of the same university.

### 2.3. Procedures

A FreeMed baropodometric platform was used to assess plantar pressure and foot geometry. All measures were processed with the integrated FreeStep software. Leg dominance was determined according to the Bogdanowicz test. This method involves ascending a step on a signal. The leg used by the participant to take the first step is treated as the dominant limb [20].

Testing was first performed in the static condition and then in the dynamic condition with no rest interval. While analysis of the feet in the static condition is considered acceptable, the literature recommends the concomitant application of dynamic testing to better reflect the functional status of the feet [21].

Both tests were performed barefoot with both eyes open. Testing in the static condition involved standing quietly for 5 s on the baropodometric platform with the feet parallel and arms resting along the body. For the dynamic condition, the participant walked at a normal pace from one end of the platform and back approximately four times to allow each foot to register on central part of the platform at least twice (FreeStep v.1.0 User’s Guide: www.koordynacja.com.pl). Variables collected in both conditions included dominant and non-dominant limb peak and average plantar pressure, forefoot and rearfoot plantar pressure distribution, the calcaneus angle as an equivalent of the gamma heel angle and plantar axis [22]. Additionally, medial and lateral plantar pressure distribution were measured in the dynamic condition. Foot posture was graded as neutral pronation, overpronation, and supination with equal load on both feet according to Myers [15].

### 2.4. Statistical Analyses

All statistical calculations were performed with SPSS 10.0 software package (IBM, Connecticut Ave NW, USA). The data set was assessed for normality using the Shapiro-Wilk test. Basic descriptive statistics (mean ± standard deviation) were calculated for all variables. Student’s *t* test was used to compare the between-group differences. Spearman’s rank correlation coefficients were calculated to determine the association between BMI and average and peak plantar pressure. Correlations were interpreted in accordance with the following scale of magnitude: 0.2 very weak correlation, 0.2–0.4 weak correlation, 0.4–06 moderate correlation, 0.6–0.8 strong correlation, 0.8–0.9 very strong correlation, and 0.9–1.0 indicates a perfect correlation. The level of significance was set to 0.05 for all procedures [23].

## 3. Results

The majority of the participants in both groups showed neutral pronation, with overpronation the least common foot posture (Figure 1). Mean calcaneus angle and plantar axis was greatest in the weightlifting group for the dominant limb in the static condition although no significant between-group differences were observed for these two variables in the static condition (Table 2). Additionally, the calcaneus angle in both groups did not meet normative values (15–18°) [21]. Plantar pressure in the static condition was greatest in both groups at the rearfoot of the dominant limb. None of the between-group differences in plantar pressure distribution in this condition were statistically significant (Table 3). Correlation analysis between BMI, peak and average plantar pressure in the static condition found only a moderately significant correlation in the weightlifting group for the non-dominant limb. No significant correlations were observed in the control group (Table 4).

Mean calcaneus angle and plantar axis measured in the dynamic condition were of greater magnitude in the group of weightlifters for the dominant limb (Table 5). The calcaneus angles in both groups did not meet published norms (normative values 15–18°) [21]. Plantar pressure distribution in both groups in the dynamic condition showed greater forefoot loading (Table 6). The difference between rearfoot and forefoot loading was smaller in the control group. However, none of the differences between the two groups were statistically significant. Correlation analysis between BMI and peak, average plantar pressure in the dynamic condition revealed no significant relationships in the group of weightlifters. However, a strong significant correlation was observed between BMI and average plantar pressure of the non-dominant limb in the control group (Table 7).

## 4. Discussion

Among professional athletes, the musculoskeletal system is subject to significant loads over the course of their career. Efforts made to minimize the risk of injury should involve frequent monitoring of the musculoskeletal system and in particular the feet, as they provide a base of support for the entire body [2]. Several researchers have suggested that static measurements of the ankle joint, neutral position of the talocalcaneal joint, and anterior and posterior measures of the feet provide sufficient data to identify specific foot postures. According to Hillstrom et al., many pathologies of the feet are biomechanical in nature and frequently associated with foot posture [24]. The main goal of the present study was to investigate the differences in static and dynamic plant pressure and other foot geometry variables between female Olympic-style weightlifters and a control group due to the overall lack of research in this area. The differences that were observed in the static and dynamic measures were not statistically significant and can be attributed to the large variation of the studied parameters in both groups. However, these findings may be important contributions towards a better understanding of the mechanisms that affect plantar biomechanics and its effects on modifying the musculoskeletal system [25,26]. Such an understanding could prevent and/or minimize the incidence of possible injuries not only among weightlifters but also for those not involved in sports. Testing in the static condition revealed smaller differences in the calcaneus angle between dominant and non-dominant limb in the weightlifting group. These values were below published norms and may be related with knee joint morphology although this was not considered in the present study. According to Kelikiana, limited extension of the knee joint and varus knee alignment can cause excessive supination and a decrease in the calcaneus angle. Consequently, this could overload the iliotibial band, cause excessive external rotation of the hip joint, lead to sacroiliac joint dysfunction, and induce changes in the zygapophyseal joints of the lumbar vertebrae [27]. Eltibi et al. found that the prompt discovery of longitudinal arch dysfunction in athletes could be crucial in the early detection of static lower limb abnormalities and prevent future disorders in this area [28]. The greatest magnitude of average plantar pressure was observed in the rearfoot of the non-dominant limb in both groups. This may be associated with the functional shortening of the superficial dorsal line, disturbed muscle tension on the plantar surface, or shortening of the hamstring muscles [15,25]. Research by Sato has shown that using weightlifting shoes can increase the angle of plantar flexion and reduce pelvic tilt, thereby transferring load to the back of the feet. This can contribute to greater activation of the knee extensor muscles, reducing shear forces as well as excess loading of the lumbar spine [29]. Akkus and co-authors examined lift technique to find that balance shifts towards the forefoot or moves backwards towards the transverse arch of the foot, and when the bar reaches knee height the knees straighten resulting in a transfer of body weight to the heels [10]. In both groups, neutral pronation was the most common foot posture and in concurrence with Gómez [30]. This may be associated with the proper development of the lateral line of the fascia among the majority of the participants. Dysfunction of this fascia line may cause excessive foot pronation or supination [15,31] Testing in the dynamic condition (when walking) showed larger differences in the calcaneus angle between the dominant and non-dominant in the weightlifting group. This may be caused by the asymmetric work of the lower limbs when performing the lunge during the jerk phase. The greater amount of forefoot loading that was observed during gait in both groups may be associated with a shortening of the anterior superficial line of the fascia [15,31]. In the static condition, only in the group of weightlifters was a moderate association found between BMI and peak and average plantar pressure of the non-dominant limb. However, in the dynamic condition, a strong association was observed between BMI and average plantar pressure of the non-dominant limb only in the control group. These findings are confirmed by Tsung, who found that excessive body weight can cause a reduction in longitudinal and transverse arch height [32]. The present studies demonstrate the need for additional functional assessments of the feet and plantar pressure using baropodometric systems. These can collect many biomechanical indicators of the feet in static and dynamic conditions [33].

## 5. Conclusions

Olympic-style weightlifting may affect plantar pressure distribution and foot posture in female weightlifters. No significant between-group differences were observed among studied variables. BMI may also correlate with peak and average plantar pressure in this population. Further research is needed to determine if Olympic-style weightlifting may affect plantar pressure distribution and foot posture in female weightlifters.

### Practical Applications

The present findings may contribute to the reduction of injuries associated with overuse of the musculoskeletal system and the occurrence of pain among individuals actively involved in sports, which can be associated with the abnormal transfer of load via the kinematic chain from the feet towards the feet, knees, hips, and spine. These findings can also be of importance for coaches and weightlifters to ensure the safe development of the musculoskeletal system and effective training programs [2,3,31,34,35].

## Figures and Tables

**Figure 1 ijerph-17-02669-f001:**
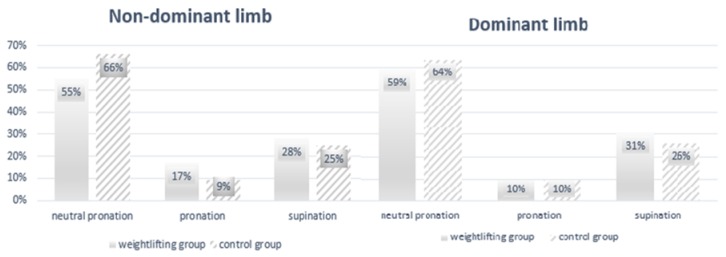
Foot type in both groups.

**Table 1 ijerph-17-02669-t001:** Anthropometric characteristics of the groups.

Variable	Weightlifting Group	Control Group	*p*
Age (years)	18.3 ± 2.6	19.3 ± 2.7	0.101
Body height (m)	163.12 ± 4.46	167.39 ± 6.04	0.008
Body mass (kg)	58.5 ± 7.61	60.78 ± 8.41	0.334
BMI (kg/m^2^)	21.96 ± 2.23	21.74 ± 3.14	0.799

**Table 2 ijerph-17-02669-t002:** Static calcaneus angle and plantar axis of the dominant and non-dominant limb and between-group comparisons (Student’s *t* and *p* value).

Variable	Weightlifting Group	Control Group	Student’s *t*	*p*
Calcaneus angle dominant limb	11.29 ± 5.25	10.87 ± 5.69	−0.26	0.793
Calcaneus angle non-dominant limb	11.04 ± 7.0	8.09 ± 4.74	−1.69	0.099
Plantar axis dominant limb	13.38 ± 5.2	11.35 ± 4.8	−1.39	0.172
Plantar axis non-dominant limb	9.20 ± 5.29	7.63 ± 5.4	−1.14	0.262

**Table 3 ijerph-17-02669-t003:** Static forefoot and rearfoot plantar pressure of the dominant and non-dominant limb and between-group comparisons (Student’s *t* and *p* value).

Variable	Weightlifting Group	Control Group	Student’s *t*	*p*
Loading of dominant limb	51.13 ± 4.67	50.52 ± 7.48	−0.03	0.714
Loading of non-dominant limb	48.88 ± 4.67	49.48 ± 7.48	0.33	0.741
Loading of forefoot of dominant limb	40.96 ± 13.34	40.13 ± 13.94	−0.21	0.836
Loading of forefoot of non-dominant limb	40.75 ± 13.74	37.04 ± 16.09	−0.85	0.399
Loading of rearfoot of dominant limb	59.04 ± 13.34	59.87 ± 13.94	0.21	0.836
Loading of rearfoot of non-dominant limb	59.25 ± 13.74	62.96 ± 16.07	0.85	0.399

**Table 4 ijerph-17-02669-t004:** Spearman’s rank correlation coefficients of BMI (Body Mass Index) and static peak and average plantar pressure of the dominant and non-dominant limb in both groups.

	BMI		Peak Plantar Pressure (gr/cm^2^)		Average Plantar Pressure (gr/cm^2^)	
			Non-Dominant Limb	Dominant Limb	Non-Dominant Limb	Dominant Limb
BMI (weightlifting group)	1.000	Spearman’s correlation coefficient	0.475	−0.70	−0.513	−0.074
		Significance	0.19	0.745	0.010	0.733
BMI (control group)	1.000	Spearman’s correlation coefficient	0.060	−0.025	0.073	0.37
		Significance	0.785	0.911	0.742	0.868

**Table 5 ijerph-17-02669-t005:** Dynamic calcaneus angle and plantar axis of the dominant and non-dominant limb and between-group comparisons (Student’s *t* and *p* value).

Variable	Weightlifting Group	Control Group	Student’s *t*	*p*
Calcaneus angle non-dominant limb	10.0 ± 5.22	9.57 ± 5.07	−0.29	0.773
Calcaneus angle dominant limb	11.5 ± 4.93	10.83 ± 4.0	−0.51	0.610
Plantar axis dominant limb	9.58 ± 4.84	7.7 ± 2.01	−1.73	0.090
Plantar axis non-dominant limb	9.96 ± 4.85	9.57 ± 3.04	−0.33	0.742

*p* ≤ 0.05.

**Table 6 ijerph-17-02669-t006:** Dynamic plant pressure distribution of the dominant and non-dominant limb and between-group comparisons (Student’s *t* and *p* value).

Variable	Weightlifting Group	Control Group	Student’s *t*	*p*
Medial loading of non-dominant limb	49.58 ± 7.61	48.09 ± 7.15	−0.69	0.491
Lateral loading of non-dominant limb	50.41 ± 7.61	51.91 ± 7.15	0.69	0.491
Medial loading of dominant limb	48.83 ± 8.2	48.35 ± 6.4	−0.23	0.823
Lateral loading of dominant limb	51.17 ± 8.2	51.65 ± 6.4	0.23	0.823
Loading of rearfoot of non-dominant limb	42.54 ± 7.00	45.91 ± 8.15	1.52	0.135
Loading of forefoot of non-dominant	57.46 ± 7.0	54.09 ± 8.15	−1.52	0.135
Loading of rearfoot of dominant limb	44.62 ± 4.95	45.39 ± 5.49	0.50	0.617
Loading of forefoot of dominant limb	55.38 ± 4.95	54.61 ± 5.49	−0.503	0.617

**Table 7 ijerph-17-02669-t007:** Spearman’s rank correlation coefficients of BMI and dynamic peak and average plantar pressure of the dominant and non-dominant limb in both groups.

	BMI		Peak Plantar Pressure (gr/cm^2^)		Average Plantar Pressure (gr/cm^2^)	
			Non-dominant limb	Dominant limb	Non-dominant limb	Dominant limb
BMI in the weightlifting group	1.000	Spearman’s correlation coefficient	0.119	0.090	0.068	−0.105
		Significance	0.579	0.674	0.753	0.624
BMI in the control group	1.000	Spearman’s correlation coefficient	0.151	−0.121	0.603	−0.037
		Significance	0.491	0.582	0.002	0.868

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
