# Peer review of "Plantar Pressure Distribution in Female Olympic-Style Weightlifters"

_ijerph, 2020, doi:10.3390/ijerph17082669_

Round 1

Reviewer 1 Report

I thank the authors for the opportunity of reviewing this work. On this regard, I think that several changes should be applied before considering this work to be published by the IJERPH. Please see the document attached.

Author Response

Thank you for the suggestion.

Best regards,

Arletta Hawrylak

Hanna Gronowska

Reviewer 2 Report

Good and impressive work, congratulations.

I recommend reworking the style of fig. 1 (lines 153-155)

Who is applying the final formating? If you as authors are involved, please look at lines 134, 160, 162

Ref 20 does not link the user guide, but the companies website, which is not really a scientific reference; furthermore I see no way of switiching the content to English language, I recommend to place this reference in the main text.

Impact of putting and releasing load may be observed as time function. You applied external measures on the foot only, I am wondering, whether there could be a diagnosis/observation technique (e.g. with 3D xrax) for looing inside the feet during load and relaxation phase.

Conclusions are a little bit to concise for my "taste", I recommend to explain your findings and assumed deductions ins some more detail.

Author Response

Thank you for suggestion.

Best regards,

Arletta Hawrylak

Hanna Gronowska

Reviewer 3 Report

The purpose of this study was to investigate plantar pressure and foot geometry variables in female Olympic-style weightlifters and age-, sex-matched control group. The authors found that no significant differences were observed in plantar pressure distribution between the two groups. They also found some correlation between BMI and foot geometry variables. Although this study was simply designed and well-conducted with an interesting topic to investigate, their analyzed results and conclusion are not enough to be published.

Here are my suggestions,

  1. I would like to recommend that the authors should emphasize the importance and significance of their findings more clearly and add a deeper explanation about the underlying mechanism into the discussion.
  2. I would like to suggest the authors merge Figure 1 and Figure 2 in order to show a clear comparison between weightlifters and controls.
  3. I think extensive editing of the English language and style is required to improve the quality of this manuscript.

Author Response

(The authors gave the same response as above.)

Round 2

Reviewer 1 Report

I would like to congratulate the authors for the performed work. I think the quality of the manuscript has significantly improved. On this regard, I have no further comments to add.

Reviewer 3 Report

Thank you for the update.